# Natural Clay Minerals as Potential Arsenic Sorbents from Contaminated Groundwater: Equilibrium and Kinetic Studies

**DOI:** 10.3390/ijerph192316292

**Published:** 2022-12-05

**Authors:** Ambrin Rehman, Shah Rukh, Samha Al Ayoubi, Seema Anjum Khattak, Ayaz Mehmood, Liaqat Ali, Ahmad Khan, Kouser Majeed Malik, Abdul Qayyum, Hikmat Salam

**Affiliations:** 1National Centre of Excellence in Geology, University of Peshawar, Peshawar 25130, Pakistan; 2College of Humanities and Sciences, Prince Sultan University, Rafha Street, Riyadh 11586, Saudi Arabia; 3Department of Soil and Climate Sciences, The University of Haripur, Haripur 22620, Pakistan; 4Institute of Soil and Environmental Sciences, Pir Mehr Ali Shah Arid Agriculture University Rawalpindi, Rawalpindi 46300, Pakistan; 5Soil and Water Testing Laboratory for Research, Rawalpindi 46300, Pakistan; 6Department of Agronomy, The University of Haripur, Haripur 22620, Pakistan

**Keywords:** sorption kinetics, isotherms studies, natural sorbents, arsenic contamination

## Abstract

Arsenic (As) contaminated groundwater is a worldwide concern due to its chronic effects on human health. The objectives of the study were to evaluate natural inexpensive raw laterite (RL) and kaolinite (RK) for their potential use as As sorbents and to understand the As sorption on laterite and kaolinite by employing sorption and kinetic models. Raw laterite and RK were tested for EC, pH, XRF and CEC as basic parameters. Batch sorption and kinetic experiments data were fitted in the sorption (Langmuir and Freundlich) model and kinetic (pseudo-first and pseudo-second order) reaction equations, respectively. Morphological and structural changes were observed in RL and RK samples before and after As saturation by employing FTIR and SEM. The major constituent in RL was Fe and Al oxides while in RK major oxides were silica and Al. The Freundlich sorption model well explained the experimental data, indicating a greater sorption capacity of RL on a hetero-layered surface compared to RK. The kinetic reaction equations showed that equilibrium was achieved after a contact time of 240 min and the adsorption was chemisorption in nature. The RL and RK were found to be effective sorbents for As removal, however, RL showed maximum As adsorption and thus superior in comparison with RK. Structural and morphological characterization reveals the role of Fe and Al oxides in the case of RL, and Al oxides in the case of RK, in the adsorption of As. Hence this study concludes that these naturally occurring inexpensive resources can be used as sorbent agents for As-contaminated drinking water treatment.

## 1. Introduction

Arsenic (As) is carcinogenic and causes severe health threats to humans [1]. It occurs naturally as a metalloid in the environment and ranges from 0.5 to 2.5 mg kg^−1^ in most of the rocks but some sediments and fine grain phosphites have high proportions [2]. Inorganic As exists as oxides of sulfur, oxygen, iron, and in trivalent (arsenite, AsIII) and pentavalent (arsenate, AsV) oxidation states [3,4]. The AsV occurrence is dominant in natural ecosystems while AsIII prevails in reduced conditions, though both are toxic and cause acute and chronic toxicity in humans [5]. 

Arsenic is released into the environment through mineral dissolution, chemical decomposition, weathering of rocks and soils and volcanic activity [6]. Minerals associated with iron (arsenopyrite (FeAsS)), sulfide ores (ruby sulfur (AsS) and orpiment (As_2_S_3_)) are major As release sources [7]. Industrial and mining processes such as dyes and colors, wood preservatives, cement works, electronics and glassware production, disinfectants and pesticides, ores and metals treatment, and cotton drying agents are also key contributors to As release into the environment [8,9,10].

Natural and anthropogenic activities have caused As contamination in groundwater [11]. The “Arsenic Problem” first came to light in the mid-1990s in Bangladesh; since then various regions of the world including Vietnam, India, China, Argentina, Chile, Mexico, Hungary and many parts of the USA have been reported to have high levels of As in drinking water [12] posing a direct threat to more than tens of millions of people across the globe [13]. About 50 to 60 million people in the Indus plain of Pakistan, with hot spots around Lahore and Hyderabad, are exposed to As-contaminated drinking water [14]. Muzaffargarh, a district of Southern Punjab, Pakistan has been reported with very high As toxicity of 906 µg L^−1^ in groundwater [15] which is beyond the permissible limit of 50 μg L^−1^ for drinking water [16]. The maximum human tolerance level in drinking water for As concentration is 10–50 μg L^−1^, however, different countries have adopted the WHO standard of 10 µg L^−1^ [17,18]. Exposure to inorganic As (>50 µg L^−1^) results in deleterious health effects [19,20] and causes arsenicosis [21], skin, lung, bladder, and kidney cancers and acute myocardial infection [22]. 

Several known treatment mechanisms for the efficient removal of As from water are coagulation [23,24,25,26,27], softening [28], anion exchange and reverse osmosis [29]. These methods have economic limitations, though, and the adsorption technique is an economic and suitable method [30] for As decontamination using laterite [31,32] and clay minerals [33]. 

Naturally occurring minerals both in the raw and modified form are widely used for the removal of inorganic and organic pollutants from contaminated water [34,35,36]. Laterite mainly consists of mineral assemblages of hematite, gibbsite, kaolinite mineral, quartz, and a small amount of goethite [37]. The adsorption mechanism of As forms by laterite and lateritic soils of India has been extensively studied [6,31,38,39,40] and their adsorption capacities were found comparable with many other synthetic adsorbents [41,42,43]. The studies showed AsIII bond formation with laterite involved both ion exchange and ligand exchange mechanisms [6]. However, in the case of laterite soil, AsIII adsorbed through physisorption and AsV through an ion exchange mechanism [40]. In another study, laterite obtained from Vietnam was used as sorbent material for AsV and concluded that the sorption process was multilayer, physical, reversible, and engaged Van Der Waals forces [44]. Clay minerals can remove organic and inorganic substances from aqueous environments [45]. Kaolinite is a 1:1 clay mineral group of alumina silicates (Al_2_O_3_(SiO_2_)_2_(H_2_O)_2_) characterized by fine particle size, chemical inertness, and a platy structure [46]. Surface-modified kaolinite with a fluoralkylsilane agent showed excellent performance of 100% As rejection even at 1000 mg L^−1^ concentration in an aqueous solution [47]. Among other cheap materials such as fly ash, dolomite and apatite powder, kaolin clay is preferred due to its appropriate pore-forming ability which is important for the fabrication of stable micro-filtration-range inorganic membranes at a lower cost [48]. The physical nature of the As adsorption process with monolayer formation of sorbate on solid surfaces of ZnO nanoparticles-coated kaolin was observed [49]. The study also revealed that As kinetic adsorption on synthesized adsorbent followed a pseudo-second-order equation, indicating internal and external mass transfer simultaneously. Kaolinite, a cheap natural material commercially available in the market, has been found effective for the removal of Pb^+2^, Ni^+2^, Cd^+2^ and Cr^+6^ ions from drinking water [50]. The metals showed multilayer adsorption on surfaces having heterogeneous binding sites and followed second-order adsorption kinetics. 

Laterite and kaolinite deposits are widely reported across the country with varied geological settings and formations [51].

Third world countries like Pakistan cannot afford expensive treatment systems or bottled water and thus need an affordable low-tech solution for safe drinking water. Identification of indigenous, low-cost clay materials has limited their use in household filter systems. Local deposits of laterite and kaolinite-rich clays can be a cheaper source for As removal from drinking water. The present study aimed to identify an effective natural indigenous raw clay material (kaolinite and/or laterite) for As removal from contaminated drinking water. The specific objectives of this study were to determine As sorption capacity of laterite and kaolinite for their potential use as sorbents and to understand the sorption kinetics of As on raw laterite and raw kaolinite. 

## 2. Materials and Methods

### 2.1. Sample Collection and Preparation

Laterite and kaolinite samples were collected from known deposits located in Khyber Pakhtunkhwa (Figure 1). Kaolinite samples were collected from the low hilly parts of Taghma, Tarkano, and the Matta area of Swat, Khyber Pakhtunkhwa. These Kaolin deposits at Swat occur within Kohistan ortho-amphibolites of Cretaceous age. The ortho-amphibolites contain felsic zones and leucocratic dikes that constitute the host rocks of Kaolin. They are part of the Cretaceous Kohistan island arc terrane [52]. The laterite samples were collected from the Kakul Tatta Pani area of Abbottabad district, Khyber Pakhtunkhwa. Laterite deposits in the Abbottabad region are hosted in the Abbottabad Formation of the Cambrian age [53,54]. A total of five different sites were sampled, namely, Tarkano-1, Tarkano-2, Taghma-1, Taghma-2 and Matta, for kaolinite, and HB-6, HB-7, HB-9, HB-10 and HB-12 for laterite. The freshly exposed surface was sampled manually by removing surface debris from the exposed deposits of kaolinite and laterite and stored in zip-lock plastic bags with proper labels. All samples were shifted to the geochemistry laboratory of NCE in Geology and were dried at room temperature. The samples were ground to pass through a 200 µm mesh size before further morphological and experimental analysis.

### 2.2. Basic Properties

The pH of each laterite and kaolinite sample was measured by making a 1:2.5 sample water suspension [55]. A twenty g sample was taken in 50 mL water, stirred and allowed to stand for 30 min. After 30 min, the pH was measured using a C931 Electrochemical analyzer made by Consort (Turnhout, Belgium) pre-calibrated with buffers 4 and 7. For EC, a 1:5 sample and water suspension was prepared [56]. A five g sample was taken in a beaker with 25 mL water, stirred and allowed to stand for 30 min before taking a reading with a C931 Electrochemical analyzer. Before taking a reading, the instrument was calibrated with a reference solution of 0.01 M KCl, with a conductivity of 1413 µS cm^−1^. 

Laterite and kaolinite clays were analyzed for cation exchange capacity (CEC) by saturating the samples with sodium cations [57]. A five g sample was taken in 50 mL centrifuge tubes and 30 mL of C_2_H_3_NaO_2_ (1M) solution was added, shaken for 5 min, followed by centrifugation at 5000× *g* rpm for 15 min, repeated three times, and the supernatant was discarded. The sample was then washed with 30 mL of 95% ethanol by shaking and centrifugation three times. Then 1 M C_2_H_7_NO_2_ (30 mL) was added to the suspension, centrifuged after 5 min shaking, repeated three times, and the decant was collected in a volumetric flask, and the volume was made up to 100 mL with C_2_H_7_NO_2_. The concentration of Na was measured in the collected supernatant using a flame photometer. The following equation was used to calculate the meq of Na/100 g.
$$\mathrm{meq}\;\mathrm{of}\;\mathrm{Na}/100\;g\;\mathrm{sample}=\mathrm{emmision}\;\mathrm{reading}\;(R\;\mathrm{meq}/L)\times \frac{100\;\mathrm{mL}}{1000\;\mathrm{mL}}\times \frac{100g}{\mathrm{Wt}.\;\mathrm{of}\;\mathrm{sample}\left(g\right)}=\frac{R\times 10}{\mathrm{Wt}.\;\mathrm{of}\;\mathrm{sample}\;\left(g\right)}$$
where R is the meq/L of Na as determined by the flame photometer. The displaced Na is the measure of the CEC of the sample. 

The elemental composition of selected laterite and kaolinite clays was determined by X-ray fluorescence. Powdered laterite and kaolinite clays were infused in pellet form with 4–5 drops of epoxy glue. An amount of 0.5 g of each of the clays was used to prepare pellet of 199 mm diameter under a 10-tonne hydraulic press. Each pellet was separately placed on the holder of the specimen chamber and bombarded with X-rays emitted from the X-ray tube window. 

### 2.3. Adsorption of As on Laterite and Kaolinite

#### 2.3.1. Batch Sorption Experiments

Batch sorption studies were carried out for sorption isotherms and the data were fitted in the Langmuir and Freundlich equations to model parameters. Triplicate 3 g samples of laterite, and kaolinite were equilibrated with 30 mL of 0.01 M KNO_3_ solution containing 0, 0.5, 2, 10, 30, 50, 70 and 100 mg L^−1^ of As using Na_2_AsO_4_. The suspension was shaken for 48 h at 25 °C and centrifuged at 5000× *g* rpm for 15 min. The supernatant was filtered through a 0.45-micron cellulose membrane, acidified and analyzed for the As solution concentration using atomic absorption equipped with hydride vapor generation (HVG-AAS). The difference in applied concentration and equilibrium As solution concentration (*C_w_*) was taken as the sorbed amount. The maximum adsorption capacity (*b*) and surface binding strength (*K*) were calculated by fitting the data in the Langmuir model (Equation (1)) [58,59,60].
(1)$$X=\frac{bKC_{w}}{1+KCw}$$

The linear form of the Langmuir model is
(2)$$\frac{C_{w}}{X/m}=\frac{1}{Kb}+\frac{C_{w}}{b}$$
where *C_w_* is the equilibrium concentration, *X* is the adsorbed concentration, *m* is the mass of soil, *K* is the surface binding strength, and *b* is the maximum adsorption capacity.

The adsorption isotherm was also fitted to the Freundlich equation [61]:(3)$$\frac{X}{m}=k_{f}C_{w}{}^{\frac{1}{\beta }}$$
or rearranged in the linear form:(4)$$\log X/m=\log k_{f}+\frac{1}{\beta }\log C_{w}$$
where *X/m* is the equilibrium concentration adsorbed by the soil (mg kg^−1^), *C_w_* is the equilibrium concentration in solution (mg L^−1^), *β* is an adsorption exponent related to adsorption intensity and *k_f_* is the Freundlich adsorption coefficient (L kg^−1^) related to maximum adsorption capacity. A plot of log *X/m* versus log *C_w_* was fitted using linear regression; *β* was found by the reciprocal of the slope of the regression line. The intercept of this regression line yielded *k_f_*.

#### 2.3.2. Adsorption Kinetics of As

In kinetic experiments, 30 mL As solution having a concentration of 0.5 mg L^−1^ was used with 3 g of laterite and kaolinite samples. The suspension was shaken at 120 rpm for specific time intervals i.e., 0, 120, 240, 480 and 720 min. Immediately, after shaking the suspension was centrifuged at 5000× *g* rpm for 5 min and the supernatant was filtered through a 0.45-micron membrane and filtrates were stored at 4 °C after acidifying till analysis. The kinetic experiment data were fitted to the Pseudo-first-order kinetic equation [62] using the Origin 8.5 software.
(5)$$q_{t}=q_{e}\;\left(1-e^{k_{1}t}\right)$$
where *q_t_* is the sorbed amount (mg g^−1^) at time t (min), *q_e_* is the sorbed amount (mg g^−1^) at equilibrium, and *k_1_* is the pseudo-first-order rate constant (min^−1^). A non-linear fit to the above-mentioned equation was employed using the Box-Lucas function which yields *q_e_* as the intercept and *k_1_* as the slope. A fitting to a pseudo-first-order kinetic model indicates that the reaction-limiting step is governed mainly by the physisorption. This model suggests that the sorption rate is proportional to the difference between the amount of adsorbed sorbate at time t and the amount of adsorbed sorbate at equilibrium. Eventually, the sorption is reversible and reaches equilibrium.

The kinetic experiment data was also fitted to the pseudo-second-order kinetic equation [63].
(6)$$q_{t}=\frac{q_{e}^{2}\cdot k_{2}\cdot t}{1+q_{e}\cdot k_{2}\cdot t}$$
where *q_t_* is the capacity (mg g^−1^) at the time (min), *q_e_* is the capacity (mg g^−1^) at equilibrium and *k_2_* is the pseudo-second-order rate constant (mg g^−1^ min^−1^). This equation was also solved by employing the Origin 8.5 software. The *q_e_* and *k*_2_ were estimated by keeping t as the independent variable and *q_t_* as the dependent variable. This model suggests that the adsorption capacity is proportional to the active sites occupied by the sorbate. A fitting to the pseudo-second-order kinetic model indicates that the reaction is governed principally by chemisorption as the limiting step.

The solution concertation for both batch sorption and kinetic adsorption experiments was maintained at 7 to mimic the real groundwater. The WHO recommends a pH range of about 6.5 or higher for drinking water [18]. Good quality water usually has pH values in the range of 6.5 and 8.5. 

After the sorption and kinetic experiments, one sample each from laterite and kaolinite was selected for the FTIR and SEM-EDX analysis. For understanding the mechanism of adsorption, the changes in each selected laterite and kaolinite sample before adsorption and after adsorption was compared.

### 2.4. Post-Adsorption Studies

#### 2.4.1. ATR-FTIR

The structural composition of laterite and kaolinite was determined by Fourier transform infrared spectroscopy (FTIR) by making a KBr pellet. Two hundred mg of the homogenized mixture of powdered sample and KBr (1:110) was pressed to form a die under vacuum. The pellet was left in a desiccator overnight and scanned between 400–4000 cm^−1^ using a Shimadzu FTIR instrument, Kyoto, Japan [64]. 

#### 2.4.2. SEM-EDX

The morphology of laterites and kaolinite was studied through scanning electron microscopy (SEM) coupled with energy dispersive X-ray (EDX) for elemental analysis. The powdered samples were gold-coated by evaporation under a high vacuum using a gold coater. 

### 2.5. Arsenic Determination

Total As concentration of the aliquots of kinetic and batch sorption experiments were determined using a Shimadzu AA-6300 atomic absorption spectrophotometer coupled with Shimadzu HVG-1 hydride vapor [65]. The absorption was measured at 193.7 nm wavelength using a slit-width of 0.7 nm, a lamp power supply of 7W, and an air/acetylene flame. In the hydride generation assembly, arsine (AsH_3_) produced by a premix of 0.4% NaBH_4_ and 0.5% NaOH solutions, was mixed with 5 M HCl in the mixing chamber [66]. The acid and the premix of NaBH_4_ and NaOH were, separately, pumped at a rate of one mL min^−1^ using a peristaltic pump to the gas–liquid separator chamber from where the AsH_3_ gas was carried to the flame by N_2_ gas at 0.32 MPa. The detection limit was 2.42 μg L^−1^ as determined by analyzing 10 blanks and calculating the standard deviation (σ = 0.26). The detection limit was a mean of 10 blank samples plus three times the standard deviation.

## 3. Results and Discussion

Raw laterite and kaolinite samples were characterized for their basic properties (pH, EC and CEC) and elemental composition using XRF. The As sorption potential of raw laterite and kaolinite was determined by batch and kinetic studies. Langmuir and Freundlich equations were used to fit the batch sorption experimental data and the models’ regression equations were employed to calculate adsorption parameters. Equilibrium data were fitted to pseudo-first- and pseudo-second-order equations. One sample each of laterite and kaolinite was selected for further morphological studies (SEM and FTIR) for understanding the sorption mechanism of As on to laterite and kaolinite. The detailed results are explained below.

### 3.1. Basic Properties

The results presented in Table 1 show that the pH of the laterite and kaolinite samples were in the acidic-to-near-neutral range. The laterite samples’ pH ranged from 5.9 to 7.1, with the lowest pH of 5.9 observed in HB-10, whereas the pH of the kaolinite samples was in the range of 6.5 to 7.5, with the highest pH observed in Taghma-1. The pH values below 7 may indicate the presence of hydrogen ions on the exchange sites which results in more anion exchange capacity. The increase in the pH values results in lower hydrogen ions on exchange sites and inflated CEC values. Our results are in agreement with Ko [67], who observed most of the laterite samples’ pH values in the acidic range. Maiti et al. [31] reported laterite pH values in the range of 7.0–7.2. Similarly, Li and Xu [68] found the pH of kaolinite in the acidic (5.3) range, which was slightly lower than the pH of the kaolinite used in this study. Moreover, most samples had pH values in the acidic range. 

The optimum pH range for the As (AsV and AsIII) adsorption on natural laterite was found in the range of 4.0 to 7.0, and in this pH range surface active sites are positively charged [6,38] while As forms exist in the form of anions (H_2_AsO_4_^−^, H_3_AsO_3_^−^) and are adsorbed onto active sites of the laterite surface. Similarly, maximum adsorption of As was observed in the pH range of 7 to 7.6. on activated alumina [69] and iron oxide-coated sand [70], as both iron and/or aluminum oxides are responsible for As adsorption on laterite. Adsorption of As on hydrous iron and/or aluminum oxide of natural laterite surface is mainly by ligand exchange, and electrostatic interaction is insignificant [71]. 

In the case of clay minerals, As adsorption on clay minerals (kaolinite, illite and montmorillonite) was found to be pH dependent [72]. It was noticed that kaolinite adsorption was maximum over the pH range of 2.0 to 5.0, and decreased steadily with an increase in pH. The presence of oxygen atoms on kaolinite surfaces forms positive charge formation in the presence of water, which is responsible for the adsorption of As forms at lower pH. However, at higher pH, kaolinite surfaces become more and more negatively charged, which prevents As adsorption. 

The EC of laterite samples varied from 3.6 to 6.7 mS cm^−1^, while kaolinite samples’ EC was observed in the range of 1.6 to 4.1 mS cm^−1^. Overall, kaolinite samples had lower mean values of EC as compared to the laterite samples. The results show a lower amount of soluble salts in kaolinite as compared to the laterite. Li and Xu [68] reported the EC of kaolinite as 10.1 mS cm^−1^, which was comparatively high compared to the results of the present study. However, Maiti et al. [39] reported much lower values of 15 μS cm^−1^ for raw laterite compared to the laterite of the present study. 

The CEC of the laterite samples was in the range of 5.42 meq/100 g to 24.95 meq/100 g, while in kaolinite, the CEC was in the range of 3.29 meq/100 g to 22.34 meq/100 g. HB-7 among laterite and Taghma-1 among kaolinite showed higher CEC values. The results in this study revealed that pH values around the neutral range (6.5 to 7.5) resulted in higher values of CEC, as was observed in both laterite and kaolinite samples, except for Taghma-1. Much lower values of Kaolinite CEC were reported by Hisseini et al. [73]; however, laterite CEC values are in line with the results of Ko [67].

The results obtained for the elemental composition of selected laterite and kaolinite samples using XRF are presented in Table 1. The elemental analysis showed that the main constituents of laterite samples are Fe_2_O_3_, Al_2_O_3_ and SiO_2_, which contributed around 24% to 58%, 5 to 13% and 0.8 to 3%, respectively. Laterites are mostly characterized by the presence of iron and aluminum oxides as major constituents as a product of intense weathering [74]. Similar values of Fe_2_O_3_ and Al_2_O_3_ for laterite derived from different geographical regions (India and Vietnam) were reported in the literature, such as Fe_2_O_3_ (51%) and Al_2_O_3_ (12.5%) [31], and Fe_2_O_3_ (49%) and Al_2_O_3_ (18%) [75]. However, lateritic soils contrarily have lower quantities of Fe_2_O_3_ [76,77]. The availability of Fe and Al compounds makes laterite a unique material that can act as a potential sorbent for As removal [31,40,78]. 

Partey [78] also reported Fe_2_O_3_, Al_2_O_3_ and SiO_2_ as major oxides in the laterites, while CaO, P_2_O_5_ and K_2_O were also present as minor components. The results of this study are in line with outcomes that reported similar values of Fe_2_O_3_ for raw laterite. On the other hand, elemental analysis of kaolinite samples showed that SiO_2_ (6–14%) and Al_2_O_3_ (5–10%) were the major oxides, as they are part of tetra and octahedral sheets, while other minor oxides are Fe_2_O_3_, CaO, P_2_O_5_, K_2_O, TiO_2_ and Mn_2_O_3_. These results confirm that this clay is an alumino-silicate material. David et al. [79] reported the presence of SiO_2_, Al_2_O_3_ and Fe_2_O_3_ in raw kaolin deposits which they related to kaolinite, quartz and hematite as major components. However, in some previous studies, significantly higher values of SiO_2_ (47–57%) and Al_2_O_3_ (22–40%) were reported for raw kaolin, compared to the present study [80,81,82].

### 3.2. Arsenic Adsorption by Laterite and Kaolinite

Adsorption isotherms for different laterite and kaolinite samples are shown in Figure 2a,b. A pronounced variation in the As adsorption behavior was observed among laterite samples, whereas kaolinite samples varied nominally, as obvious from the rise of the isotherms below the equilibrium concentration of 20 mg L^−1^, both for laterite and kaolinite. Generally, laterite isotherms indicated a higher rate of adsorption (up to ~800 mg kg^−1^) compared to the kaolinite, which achieved adsorbed concentration of ~600 mg kg^−1^. In Figure 2a, it can be observed that laterite sample HB-7 has not reached the maxima and it can adsorb more than 1000 mg of As per kg of adsorbent. The remaining laterite samples’ isotherms indicated that they can adsorb ~800 mg of As per kg of adsorbent used. In a study, raw laterite collected from India showed notably lower AsV and AsIII adsorption (100 to 120 mg kg^−1^) [31]. Natural laterite collected from Vietnam also showed slightly lower adsorption of AsV and AsIII (400 to 600 mg kg^−1^) compared to the present study [75]. However, acid–base treated laterite on the other hand showed much higher adsorption (up to ~4000 mg kg^−^1) for AsV and AsIII [31,39]. Similarly modified kaolinite also showed higher adsorption of As (2000 to 2500 mg kg^−1^) compared to materials used in this study [82].

The fast increase in the sorption of As in the case of laterite was observed as sorbed concentration reached ~300 mg kg^−1^ with a small increase (~3 mg L^−1^) in the equilibrium concentration. Similar behavior of the As isotherm, as in the fast initial rise and moderate increase in the lateral section of the isotherm, was reported by previous studies [39,75]. A higher rate of adsorption in the laterite in comparison with kaolinite may be correlated with a higher amount of iron and aluminum oxides in laterites. The role of iron and aluminum oxides in laterite for As sorption is already well explained in the literature [31,75].

The Langmuir and Freundlich sorption models were then used to explain the sorption isotherms data. The data of the present study fitted well in the Freundlich sorption model, and the Langmuir model was unable to fit the sorption data both for laterite and kaolinite samples. The Freundlich isotherm can be theoretically derived by assuming that the adsorbent surface is heterogeneous, that is, the adsorption sites are grouped in one patch and distributed exponentially to the adsorption energy. In addition, the adsorption energy usually differs between patches. The Freundlich isotherm gave a better description of the equilibrium behavior, indicating that the sorbents used in this study had a heterogeneous surface, and adsorption energy, which is formed by the interaction between the adsorbate and the adsorbent, was distributed following the exponential decay function. That is why only the results of the Freundlich model parameters are given below.

### 3.3. Freundlich Model Parameters

Data for As sorption on different laterite and kaolinite types fitted well in the Freundlich equation (Figure 2c,d). The coefficient of regression (R^2^) values for laterite were in the range of 0.93 to 0.99, and for kaolinite were in the range of 0.94 to 0.99. The adsorption parameters obtained from the linear regression equation are presented in Table 2. The adsorption intensity, *β*, was in the range of 0.46 to 1.16 for laterite, and kaolinite was in the range of 0.78 to 0.98, while relative adsorption capacity, *k_f_*, was in the range of 8.52 L kg^−1^ to 1191 L kg^−1^, and kaolinite ranged from 9.12 L kg^−1^ to 28 L kg^−1^. Overall, it seems that kaolinite had a slightly higher adsorption intensity for As than laterite, except for HB-9, which showed the highest (1.16) adsorption intensity, and laterite had more adsorption capacity compared to kaolinite, with an exceptional high of 1191 L kg^−1^ for HB-7. In the Freundlich isotherm, the value of *β* was >1 in most laterite and kaolinite samples, indicating chemical adsorption on the heterogenous surfaces of the sorbent [83,84]. The applicability of Freundlich isotherm in the present study allowed for predicting that multilayer adsorption took place, which is in line with the results of Sanou et al. [44]. The chemisorption route of As adsorption was also observed for laterite and bentonite derived from India [77].

Higher adsorption capacities for laterite are probably due to higher levels of iron and aluminum oxides, as a positive correlation of adsorption capacity, *k_f_*, was observed with Fe_2_O_3_ (r 0.3) and Al_2_O_3_ (r 0.4). In this study, the Freundlich model was well fitted to the batch experiment data, which is in agreement with the findings of Maji et al. [40] and Nguyen et al. [83]. Also, Maji et al. [74] used the Freundlich model for laterite soils and found lower values for adsorption capacity and higher adsorption intensity values compared to the present study.

Kaolinite had significantly lower values for Fe_2_O_3_, but the adsorption of As in kaolinite might be contributed by Al_2_O_3_ and TiO_2_. Glocheux et al. [85] also reported the role of Al_2_O_3_ and TiO_2_ in the adsorption of AsIII and AsV. Almost similar values of adsorption capacity and adsorption intensity for kaolinite-supported nanocomposites were observed by fitting the Freundlich adsorption model [49]. Mudzielwana et al. [82] reported an adsorption capacity of 2.3 mg g^−1^ for AsIII, and 2.88 mg g^−1^ for AsV, for surfactant-modified kaolin clay mineral, and found that surfactant-modified kaolin clay showed good adsorption capacity towards As, as compared to unmodified kaolin clay mineral.

Mohapatra et al. [72] compared kaolinite with illite and montmorillonite as adsorbates for As removal. The Freundlich adsorption parameters showed kaolinite is a better adsorber for As than illite and montmorillonite.

Based on the above results, it could be concluded that raw laterite and kaolinite are potential adsorbents for As removal from aqueous solution. The adsorption capacities of both adsorbents are comparable with acid–base-treated laterite, and kaolinite modified with hexadecyl-trimethylammonium bromide (HDTMA-Br) cationic surfactant, in previous studies [31,82].

### 3.4. Adsorption Kinetics

To find out the efficiency of the As adsorption mechanism, optimum contact time for As uptake as well as the rate-limiting step, two kinetic models, viz., a pseudo-first-order model based on physisorption and a pseudo-second-order reaction model based on chemisorption, were analyzed. The initial concentration of As was fixed at 0.5 mg L^−1^ and the adsorbent dose was 30 g L^−1^. The shaking time was varied in the range of 0–12 h. It was observed that As removal was rapid within 120 min of contact time and relatively slower at contact beyond 120 min. This rapid removal below 120 min indicates the greater number of available sites for the As on the surface of the adsorbent. However, 4 h contact time was sufficient to get the maximum As adsorption with continuous agitation.

Pseudo-first-order and pseudo-second-order reaction models are expressed in Equations (5) and (6). The nonlinear plot of As adsorbed amount, *q_t_* (mg g^−1^), versus variable time intervals is shown in Figure 3. The parameters for pseudo-first-order and pseudo-second-order models are presented in Table 3.

In the case of pseudo-first order, the sorbed amount at equilibrium, *q_e_*, for laterite is in the range of 0.0037 to 0.0053 mg g^−1^, while for kaolinite it is in the range of 0.0016–0.0119 mg g^−1^. Overall, laterite showed a higher sorbed amount at equilibrium compared to kaolinite, except for Taghma-2. The values of the pseudo-first-order rate constant *k_1_* for laterite are in the range of 0.0032–0.02 min^−1^, and for kaolinite are in the range of 0.0027–2.48 min^−1^. Similarly, the pseudo-second-order rate constant *k_2_* for laterite is in the range of 0.3749–3.3890 g mg^−1^ min^−1^, and for kaolinite is in the range of 0.0027–2.48 min^−1^. Arsenic adsorption kinetics of clay minerals (nontronite, montmorillonite and kaolinite) showed higher *q_e_* values (0.179 to 0.267 mg g^−1^) and lower *k_1_* (0.001 to 0.0032 min^−1^) and *k_2_* (0.0191 to 0.066 g mg^−1^ min^−1^) values compared to the materials used in this study [86]. In another study, iron oxide-coated rock was used for As adsorption, in which *k_2_* and *q_e_* were 0.033 g mg^−1^ min^−1^ and 4.18 mg g^−1^, respectively [74]. Arsenic species adsorption (AsV and AsIII) on surfactant-modified kaolinite showed *q_e_* values in the range of 2.1 to 4 mg g^−1^, and *k*_1_ and *k*_2_ values in the range of 0.09 to 0.1 and 0.03 g mg^−1^ min^−1^ to 0.27 g mg^−1^ min^−1^, respectively [82].

The pseudo-second-order parameters, viz., *q_e_* and *k_2_* values, are greater than pseudo-first-order reaction parameters, and *k_2_* is many folds higher as compared to *k_1_*. Overall, *q_e_* and *k_2_* values for laterite are greater than kaolinite values except for Taghma-2 which had exceptionally higher values of *q_e_*, which were similar to the *q_e_* values of pseudo-first-order parameters.

Ghorbanzadeh et al. [86] studied the As adsorption on different clay minerals, and kinetic studies of As forms showed that the sorption mechanism was well explained by a pseudo-second-order model, and the results of pseudo-first-order parameters were in line with the present study. The kinetic behavior of As adsorption on laterite soil in an aqueous medium was studied by Maji et al. [40], employing both pseudo-first-order and pseudo-second-order kinetic models, and found that a pseudo-second-order kinetic model best described the adsorption process. Similarly, Maiti et al. [38] also reported that As adsorption on laterite was well explained by a pseudo-second-order reaction. In another study, laterite was used for the removal of As from groundwater. The kinetic modeling in the study, by considering the pattern of the plots and comparing the *q_e_* values qualitatively, concluded that the pseudo-second-order model better describes the behavior of As adsorption by laterite [87].

In the present study, the correlation coefficients (R^2^) for the pseudo-first- and pseudo-second-order equations are satisfactory (Table 3), which means that the sorption follows both equilibrium models fairly well, possibly because total As contains both AsIII and AsV, and in the pH range of 5 to 7 both AsIII and AsV can exist, which results in the sorption following a mixed model [82]. However, the R^2^ of the pseudo-second-order equation is better than the pseudo-first order’s for most laterite and kaolinite samples, suggesting the chemisorption of As on clay materials as a predominant mechanism [82]. Ho and Mckay [88] stated that the adsorption mechanism is mainly chemisorption when it follows the pseudo-second-order kinetic model. According to the pseudo-second-order adsorption rate constant *k_2_*_,_ it can be concluded that the adsorption process on laterite reached a faster equilibrium than that of kaolinite.

Thus, from the present kinetic reaction study, it may be concluded that the sorption of As on laterite and kaolinite can be better explained by the pseudo-second-order kinetic model than the pseudo-first-order kinetic model.

### 3.5. FTIR Spectra

FTIR spectra of raw laterite and kaolinite before and after adsorption are depicted in Figure 4 and Figure 5. For laterite, in Figure 4, it was observed that there is a broad stretch at 3200 to 3400 cm^−1^ with a peak at 3330 cm^−1^, confirming the presence of the hydroxyl group (bonded-OH stretch) in the post adsorption sample, as the range between 3370–3405 cm^−1^ is designated for OH stretch [89]. The broad bands around 3330 and 1634 cm^−1^ in post-adsorption samples were assigned to adsorbed water, as a range of around 3365 and 1635 cm^−1^ for adsorbed water was also reported by Mbaye et al. [90]. Separately, a range of 1620–1640 cm^−1^ responsible for interlayer water molecules between the adsorbent layer was also reported by Saadon et al. [89]. The sharp peaks at 980 and 528 cm^−1^ confirmed the presence of Al-OH and Fe-O bond stretching with slight variation in frequency (912 and 543 cm^−1^) reported by Saadon et al. [89]. Maiti et al. [31] related the peaks of 535 and 472 cm^−1^ with the presence of hematite in laterite.

In the FTIR patterns of kaolinite (Figure 5), the bands range between 3620, 3690 and 3665 cm^−1^ in the case of the post-adsorption kaolinite sample, representing the -OH stretching vibration of the adsorbed water molecule, as the range between 3669 and 3696 cm^−1^ is designated for OH stretch [90]. Similarly, the bands located at 3696, 3669 and 3652 cm^−1^ are the typical signature of the inner-surface hydroxyls of kaolinite, and the peak at 3619 cm^−1^ is attributed to the stretching frequency of the internal hydroxyl groups [91]. The broad bands around 3365 and 1635 cm^−1^ are due to the adsorbed water in post-adsorption sample of kaolinite. The bands with a slight variation at 1028, 910, 1005 and 748 cm^−1^ represent the vibration of Si-O-Fe, Si-O, Al-OH, and Fe-OH, respectively, as the values of 1034, 1008, 911, and 795 cm^−1^ assigned to these groups [31]. Ioannou et al. [92] also reported almost similar band values frequency for kaolinite–hematite composite. The peak value at 910 is attributed to the Al-OH bond stretching, as Maiti et al. [39] reported 912 cm^−1^ for this group. Moreover, the bands at 534 and 465 cm^−1^ in kaolinite can be ascribed to the vibration of Si-O-Al/Fe, Si-O-Si and Al-O, as the range of 573 and 473cm^−1^ has been studied in previous studies [93]. The specific peak at 534 is attributed to the Al-O bending vibration [94]. The peaks at 748 and 465 are related to the presence of quartz as Mustapha et al. [95] attributed 750 and 467 cm^−1^ to quartz. The band at 1114 cm^−1^ corresponds to the stretching vibrations of the Si-O-Si group, as described by Mbaye et al. [90] in the range of 1100cm^−1^. Jalees et al. [50] reported almost similar bands with slight variation in the frequency for a local kaolin called chikini mitti, obtained from the market. All the peaks in laterite and kaolinite were more intensified after adsorption.

### 3.6. Degree of Lateritization

Laterite composition widely varies with extent of lateritization, parent rock and geographical location [96]. Lateritization by hydrolysis and oxidation removes silica from the parent rock, leading to the formation of laterite concretion or laterite soil. The degree of lateritization is defined as the silica–sesquioxide (S–S) ratio. The degree of lateritization is estimated by using the expression: (SiO_2_/(Fe_2_O_3_ + Al_2_O_3_)). The ratios for all raw laterite samples used in this study were calculated from chemical composition data obtained from XRF results. The values of S–S for laterite used in the present study were in the range of 0.04, 0.06, 0.04, 0.047 and 0.015. According to Partey [78], all laterite samples with S–S values below 1.33 are characterized as laterite. All collected samples are classified as laterite or laterite concretion according to categories developed based on the degree of lateralization [31]. The information on As removal efficiency compared with different laterite compositions collected from different geographical locations is scarce. In this study laterite samples collected from different locations showed a high degree of lateritization which means they have a higher percentage of iron and aluminum oxides compared to the raw laterite used for As removal by Maiti et al. [31]. The iron and aluminum oxide percentage in laterite is important for its As removal efficiency.

### 3.7. SEM-EDX

The morphology of selected laterite and kaolinite samples before and after sorption are shown in Figure 6a–d, along with the EDX spectra in Figure 6e–h. The SEM images revealed that the surfaces of the laterite and kaolinite are rough, along with particles having irregular shapes and sizes. The presence of a large number of such particles may attribute to the nanostructures and magnetic properties [97], which contribute to the tendency for agglomeration and aggregation. The SEM images of laterite and kaolinite before and after sorption reveal clear changes in the morphology, which can be attributed to the adsorption of As in the pores and the roughness of the natural laterite and kaolinite surface. It can be seen that laterite and kaolinite have sheet-like morphology, which indicates the layered structure of the sorbents. The EDX spectrum of the laterite before and after sorption shows peaks of Fe and Al, which confirms the presence of Fe and Al oxides as a major constituent. Similarly, the EDX spectrum of kaolinite before and after sorption shows the peaks of Al and Si as major constituents, while Fe is in much lesser quantities. The EDX results of laterite and kaolinite are in line with the data of XRF. Hence, the role of Fe and Al oxides in the case of laterite, and Al oxides in the case of kaolinite, is clear. Based on the results, it is very much evident that the surface complexation reaction has occurred between aqueous As and sorbents (laterite and kaolinite). Thus, the sorbents used in this study have proven to be one of the most efficient sorbents for aqueous As remediation.

## 4. Conclusions

In the present study, indigenous deposits of laterite and kaolinite were studied for their adsorption potential for As, and for understanding the mechanism of As adsorption. The morphological and structural analysis (FTIR and SEM-EDX) are in agreement with the XRF analysis, and confirm the role of Fe and Al oxides in As sorption onto laterite and kaolinite. Among Langmuir and Freundlich sorption models, the Freundlich isotherm model well explained the sorption of As on the heterogenous surfaces of the sorbents. The sorption parameters indicated that laterite had greater adsorption capacity than kaolinite, while both showed almost similar intensities for As. Also, a pseudo-second-order equation better fitted the experimental data, suggesting that both physical and chemical sorption contributed to the overall sorption of As onto laterite and kaolinite, but chemical sorption contributed more, especially in the case of kaolinite. Experimental results demonstrated that naturally occurring laterite and kaolinite are quite effective sorbents for the removal of As from an aqueous solution. It has also been observed that laterite exhibited the maximum removal of As and thus proved to be superior in comparison with kaolinite. Thus, these naturally existing substances may have some applied significance as sorbent agents at the domestic level to provide safe consumable water.

## Figures and Tables

**Figure 1 ijerph-19-16292-f001:**
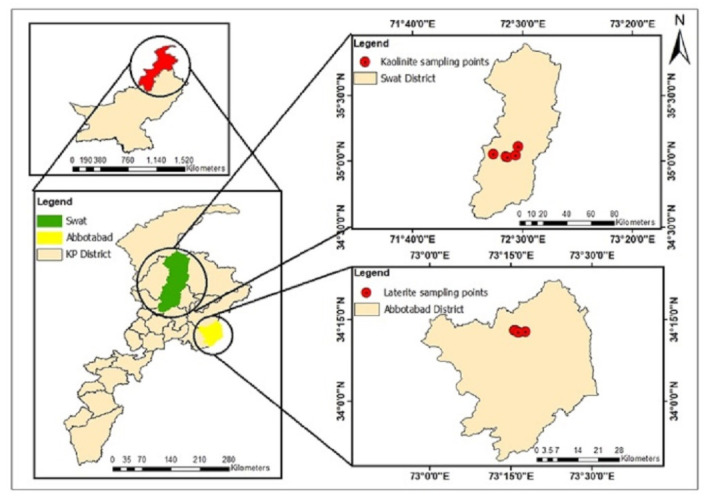
Map showing the study area and Kaolinite and Laterite sampling points in the Swat and Abbottabad districts, Khyber Pakhtunkhwa, Pakistan.

**Figure 2 ijerph-19-16292-f002:**
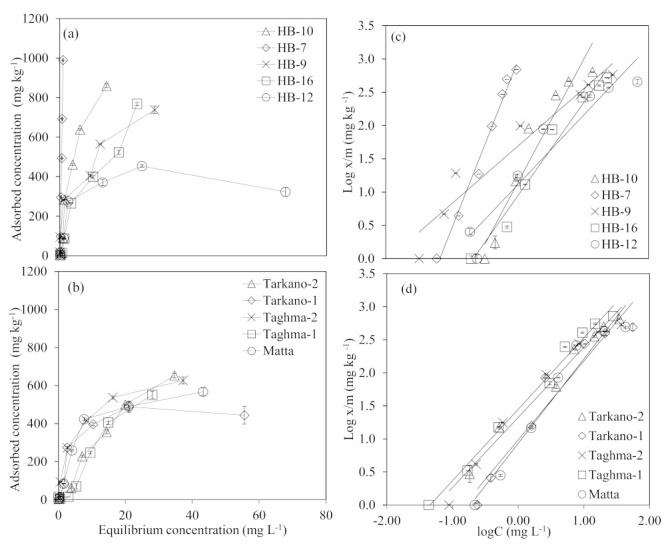
Adsorption isotherms of (**a**) laterite and (**b**) kaolinite, (**c**,**d**) depicts the Freundlich Equation (4) fit for As in the laterite and kaolinite, respectively. The trend line equation was solved to get adsorption parameters. Error bars indicate the standard deviation among replicates (*n* = 3).

**Figure 3 ijerph-19-16292-f003:**
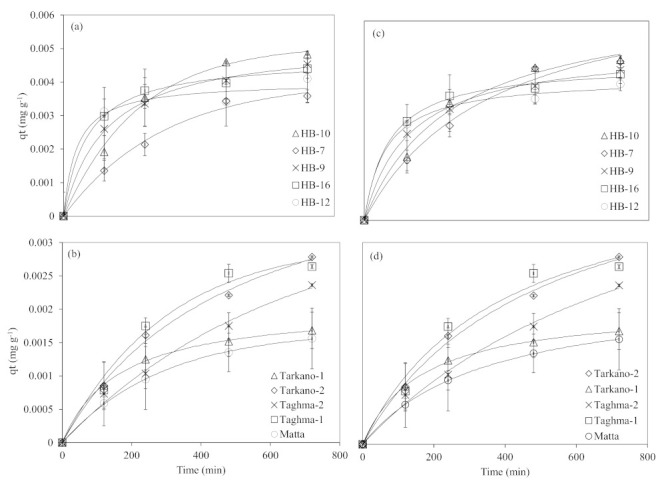
The pseudo-first-order equilibrium model fits for (**a**) laterite and (**b**) kaolinite and pseudo-second-order equilibrium model fits for (**c**) laterite and (**d**) kaolinite.

**Figure 4 ijerph-19-16292-f004:**
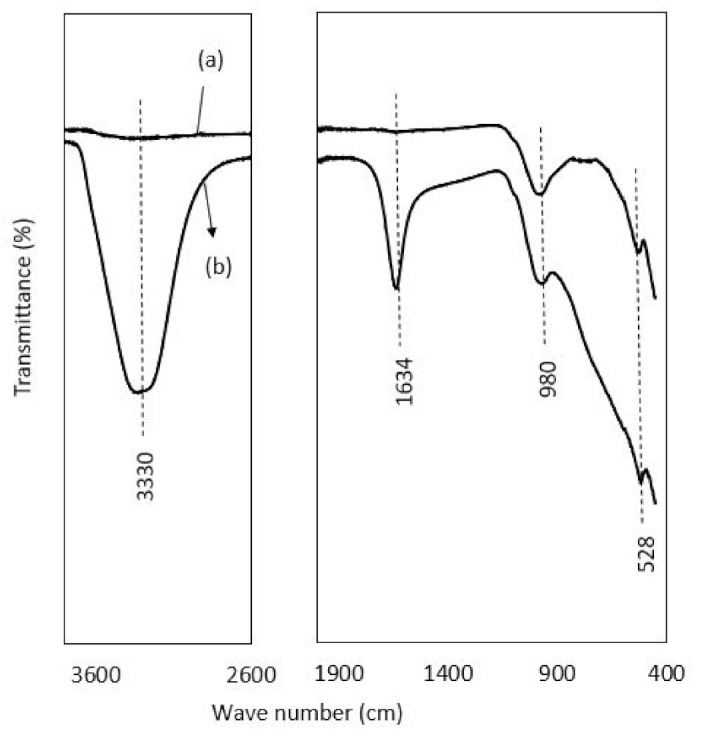
FTIR spectra showing the peak values of transmittance against wave numbers: (**a**) represents the spectra of raw laterite, and (**b**) is the spectra of laterite after adsorption.

**Figure 5 ijerph-19-16292-f005:**
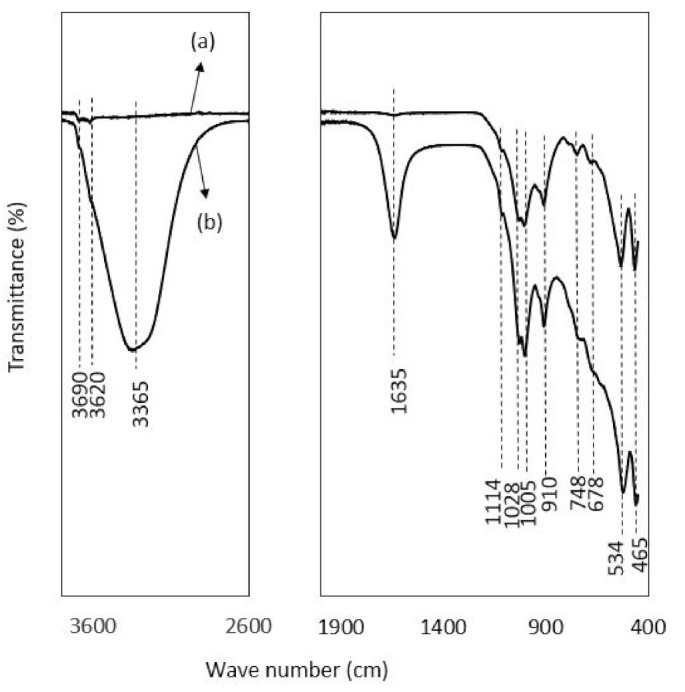
FTIR spectra showing the peak values of transmittance against wave numbers: (**a**) represents the spectra of raw kaolinite, and (**b**) is the spectra of laterite after adsorption.

**Figure 6 ijerph-19-16292-f006:**
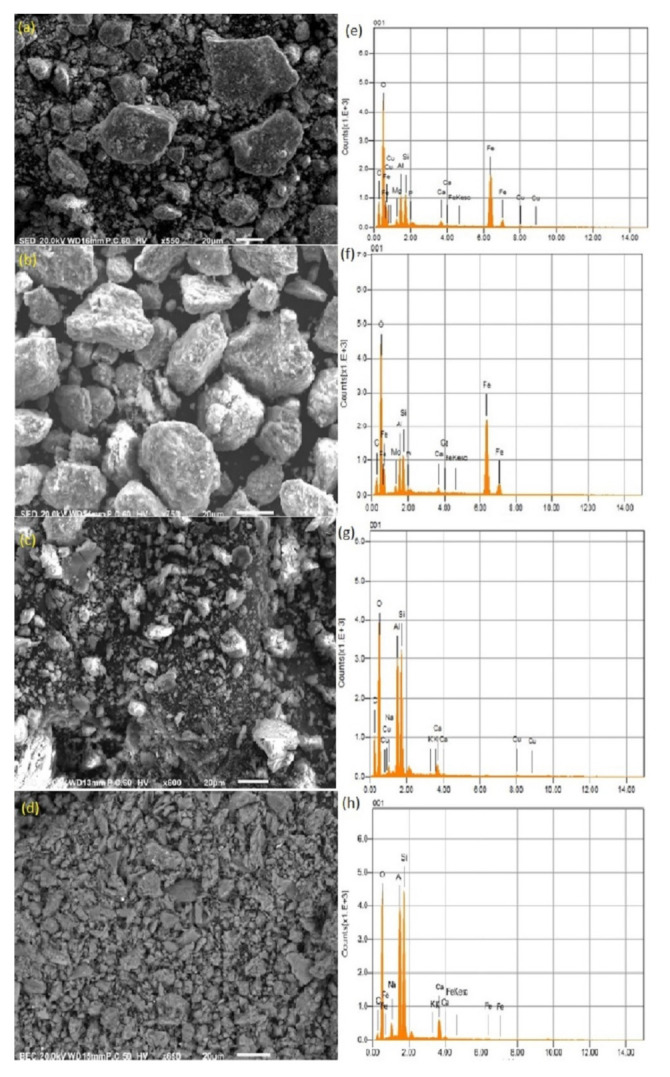
SEM micrographs along with EDX plots of a selected laterite (**a**,**b**,**e**,**f**) and kaolinite (**c**,**d**,**g**,**h**) samples taken before and after sorption.

**Table 1 ijerph-19-16292-t001:** Characteristics of the raw laterite and kaolinite samples.

Sample	Fe_2_O_3_	Al_2_O_3_	SiO_2_	CaO	K_2_O	P_2_O_5_	TiO_2_	Mn_2_O_3_	pH	EC	CEC
	%	%	%	%	%	%				mS/cm	meq/100 g
Laterite											
HB-10	46.27	5.94	2.22	1.58	0.027	6.44	-	0.11	5.9_(0.5)_	3.6_(0.8)_	7.62_(2.45)_
HB-7	45.33	6.38	3.18	3.10	0.02	1.51	-	0.1	6.5_(0.4)_	4.8_(1.3)_	24.95_(2.44)_
HB-9	24.44	12.85	1.69	0.4	0.012	0.41	-	0.05	7.1_(0.4)_	6.2_(0.7)_	15.41_(6.12)_
HB-16	58.14	6.4	3.09	3.07	0.03	0.52	-	0.12	7.1_(0.3)_	6.2_(1.0)_	29.27_(3.67)_
HB-12	49.74	4.47	0.84	20.08	0.16	0.73	-	0.14	6.2_(0.6)_	6.7_(1.2)_	5.42_(1.22)_
Kaolinite											
Tarkano-2	0.6	5.14	10.78	4.71	0.05	0.2	0.2	0.84	6.7_(0.6)_	2.6_(0.6)_	14.55_(2.45)_
Tarkano-1	0.06	6.7	11.52	5.56	0.06	0.13	1.3	0.6	7.1_(0.3)_	1.6_(0.5)_	12.81_(2.45)_
Taghma-2	1.07	9.77	13.91	5.07	0.1	0.13	0.13	0.02	6.9_(0.3)_	3.2_(0.3)_	22.34_(6.12)_
Taghma-1	0.34	3.88	8.01	5.41	0.13	0.2	0.2	0.5	7.5_(0.2)_	4.1_(0.3)_	3.29_(1.22)_
Matta	1.48	5.62	10.42	7.56	0.14	0.11	0.11	0.5	6.5_(0.5)_	2.1_(0.4)_	5.02_(1.23)_

Values in parenthesis indicate standard deviation (*n* = 3).

**Table 2 ijerph-19-16292-t002:** Fitted Freundlich adsorption model parameters for laterite and kaolinite.

Sample	*β*	*k_f_*	R^2^
		L kg^−1^	
Laterite			
HB-10	0.74_(0.29)_	13.71_(2.76)_	0.93
HB-7	0.46_(0.06)_	1191_(272.0)_	0.99
HB-9	1.16_(0.007)_	48.10_(1.83)_	0.95
HB-16	0.71_(0.018)_	8.52_(0.72)_	0.98
HB-12	0.97_(0.09)_	12.99_(3.52)_	0.93
Kaolinite			
Tarkano-2	0.98_(0.10)_	22.09_(7.68)_	0.99
Tarkano-1	0.85_(0.05)_	9.84_(2.07)_	0.94
Taghma-2	0.94_(0.02)_	21.13_(1.86)_	0.98
Taghma-1	0.93_(0.01)_	28.04_(0.66)_	0.99
Matta	0.78_(0.01)_	9.12_(0.47)_	0.97

**Table 3 ijerph-19-16292-t003:** Pseudo-first-order and pseudo-second-order reaction model parameters.

Samples	Pseudo-First Order			Pseudo-Second Order		
	*q_e_*	*k* _1_	R^2^	*q_e_*	*k* _2_	R^2^
	mg g^−1^	min^−1^		mg g^−1^	g mg^−1^ min^−1^	
Laterite						
HB10	0.0053_(0.0004)_	0.0046_(0.0023)_	0.990	0.0063_(0.0012)_	0.6218_(0.7977)_	0.976
HB-7	0.0053_(0.0003)_	0.0032_(0.0001)_	0.980	0.0074_(0.0005)_	0.3749_(0.0779)_	0.973
HB-9	0.0045_(0.0008)_	0.0085_(0.0046)_	0.966	0.0055_(0.0017)_	2.6543_(2.1387)_	0.973
HB-16	0.0042_(0.0001)_	0.0098_(0.0012)_	0.980	0.0047_(0.0002)_	3.1406_(0.2822)_	0.986
HB-12	0.0037_(0.0001)_	0.0205_(0.0167)_	0.946	0.0042_(0.0002)_	3.8895_(0.8504)_	0.990
Kaolinite						
Tarkano-2	0.0032_(0.0002)_	0.0027_(0.0003)_	0.983	0.0046_(0.0004)_	0.4446_(0.1197)_	0.990
Tarkano-1	0.0016_(0.0002)_	0.0058_(0.002)_	0.986	0.0020_(0.0001)_	3.0504_(1.1881)_	0.990
Taghma-2	0.0119_(0.010)_	0.0043_(0.003)_	0.983	0.1582_(0.24)_	1.4279_(1.3009)_	0.990
Taghma-1	0.0030_(0.0001)_	0.0033_(0.0002)_	0.973	0.0042_(0.0002)_	0.6053_(0.0878)_	0.966
Matta	0.0017_(0.0006)_	0.0036_(0.0013)_	0.990	0.0024_(0.001)_	1.7209_(1.4572)_	0.990

## Data Availability

The data presented in this study are available on request from the corresponding author.

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
