# Peer review of "Natural Clay Minerals as Potential Arsenic Sorbents from Contaminated Groundwater: Equilibrium and Kinetic Studies"

_ijerph, 2022, doi:10.3390/ijerph192316292_

Round 1

Reviewer 1 Report

Interesting project relating to the evaluation of the potential use of natural inexpensive raw laterite (RL) and kaolinite (RK) as As sorbents from aqueous sources.

However a few issues arise from the analysis of the paper:

Review carefully the English language throughout the paper.

Normalize the formatting of the equations and the used symbols in the equations and in the text (especially what should be equation 1). Equation 5 needs to be corrected, the exponential function is not presented properly.

Correct the arsine chemical formula. Revise and detect the typos.

I would be interesting to have an idea of the level of contaminant removal provided by the adsorbents in order to complement the more theoretical approach regarding the equilibrium and kinetics analysis of the adsorption process.

In Figure 3 specify more clearly what plots (c) and (d) represent. In the Figure 3 caption there is a reference to the Figure itself. All references to Tables and Figures should be in the text.

It would be useful to have a comparison of the values from Table 3 with alternative adsorbent materials.

Improve the visibility of Figure 6 (especially the plots).

Reviewer 2 Report

This paper is well written and well organized. Anyway, regarding the objective of this study, some specific improvements should be considered:

1.      In the abstract, line 25-26, it was described about material HB-12 and Tarkano-1, which the reader doesn’t know which type of materials are, please modify or identify clearly. and 3.05 g mg−1 min−1.

2.      Table 1. Characteristics of the raw laterite and kaolinite samples, please identify the number in the parenthesis.

3.      In section 3.2. Arsenic Adsorption by Laterite and Kaolinite, due to both of adsorbents, have been studied for As adsorption from various researches. Could you please compare your adsorption capacities with others?

4.      The title of Figure 6 should identify EDX analysis. The figures for EDX should increase the size of chemical elements.

5.      The objective of this study is to identify an effective natural indigenous raw clay material (kaolinite and/or laterite) for As removal from contaminated drinking water. Therefore, the conclusion should be compared which adsorbent is better and standard of drinking water.

Reviewer 3 Report

the authors describes the use of Raw laterite and kaolinite for the Batch sorption of Arsenic and kinetic experiments data were fitted in the sorption (Freundlich) model and kinetic (pseudo-second order). The author did not describe the experimental max adsorption capacities for each of the Raw laterite and kaolinite.  section 3.1. Basic Properties should be the optimization section, and each effect of parameter such as pH of samples, the pH of the working solution etc should be described properly. 

3.6. Degree of Lateritization: what is usefulness of the this degree? should be discussed properly. 

The following references should also be included. 

Acta Chimica Slovenica 64(2) 2017, 449-460.

Clean-Soil Air Water. 42(11) (2014) 1500-1508

Colloids and Surfaces A: Physicochemical and Engineering Aspects,
Volume 602, 2020, 125060, https://doi.org/10.1016/j.colsurfa.2020.125060.

Round 2

Reviewer 1 Report

In the revised manuscript, the authors address satisfactorily the recommendations received. I appreciate the effort, and now, I invite the authors to check their work carefully before resubmission, particularly, in taking special consideration to the English proofreading that has been carried out. I think that after this step the manuscript will be ready for acceptance.